# Time-Dependent Rheological Properties of Cemented Aeolian Sand-Fly Ash Backfill Vary with Particles Size and Plasticizer

**DOI:** 10.3390/ma16155295

**Published:** 2023-07-27

**Authors:** Baogui Yang, Zhijun Zheng, Junyu Jin, Xiaolong Wang

**Affiliations:** 1School of Energy and Mining Engineering, China University of Mining and Technology-Beijing, D11 Xueyuan Road, Haidian District, Beijing 100083, China; 2State Key Laboratory of Coal Resources and Safe Mining, China University of Mining and Technology-Beijing, D11 Xueyuan Road, Haidian District, Beijing 100083, China

**Keywords:** cemented aeolian sand-fly ash backfill, aeolian sand and coal gangue-filled slurry, rheological, plasticizer, particle gradation

## Abstract

The use of cemented Aeolian sand-fly ash backfill (CAFB) material to fill the mining area to improve the surface subsidence damage caused by underground coal mining is in the development stage. Their performance with large overflow water and strength loss is not well understood. Few research has been conducted to understand the effects of aeolian sand and coal gangue on the rheological properties of CAFB with plasticizers. Therefore, this study aims to investigate the effects of a plasticizer on the rheological properties, specifically yield stress and viscosity, of CAFB prepared with aeolian sand and coal gangue. CAFB mixes containing 0%, 0.05%, and 0.1% plasticizers were prepared, and yield stress and viscosity were determined at different intervals. Additional tests, such as thermal analysis and zeta potential analysis, were also conducted. It was found that the rheological properties of CAFB are the comprehensive manifestation of the composite characteristics of various models. Reasonable particle size distribution and less plasticizer can ensure the stability of the slurry structure and reduce the slurry settlement and the risk of pipe blocking. The findings of this study will be beneficial in the design and production of CAFB material.

## 1. Introduction

In China, the use of cemented Aeolian sand-fly ash backfill (CAFB) material to improve the surface subsidence and environmental damage caused by coal mining is in its embryonic stage [1,2,3]. Yushen mining area, located in the hinterland of the famous Yushenfu-Dongsheng coalfield, known as one of the world’s seven big coalfields, has hundreds of coal mines of various sizes distributed in the Yushen mining area, which are located in the Maowusu Desert and have harsh geological conditions [4,5,6]. It can reduce the damage to the surface environment caused by large-scale coal mining, which conveys a large amount of covering aeolian sand on the surface to the mining area through pipeline pressurized [7,8,9]. CAFB material is usually considered a high-density, non-segregating mixture containing 68–75% solids weight [10,11,12,13]. A hydraulic binder such as cement, in amounts of 8–12% of the total CAFB material, is an integral part of any CAFB material, which gradually adds additional bonding and mechanical strength during the maintenance process, where increasing the solids content of the CAFB mix is more economical than increasing the amount of binder, which accounts for 75–80% of the cost of the filling operation [14,15]. Generally, fresh CAFB is transported by gravity or a pipeline pumping system to the downhole mining area. However, in the transportation process, CAFB, with a very low water/ash ratio (W/A), may cause huge frictional resistance and pipeline blockage due to its rheological properties changing with the hydration age. Using a single aggregate of aeolian sand in the CAFB slurry will cause large bleeding in the underground mining area, worsening the underground operating environment. The loss of the cement, fly ash and water-mixed slurry will be reduced the strength of the filling body. The author innovatively proposes the high-concentration filling cementing material CAFB, which is prepared by mixing aeolian sand, coal gangue, cement, fly ash and water in a certain ratio, studying the influence of coal gangue dosage and hydration time on the rheology of CAFB slurry, optimizing the material ratio, reducing the frictional resistance of slurry in the pipe, realizing efficient transportation, and improving the underground operating environment and improving the strength of the filling body [2,3,7,8,16]. The yield stress and viscosity are the key rheological parameters for evaluating the CAFB material transport capacity in the design of pipeline reticulation systems. The yield stress is typically realized as the effect of conditions of mutual attraction between individual particles, which can aggregate to form suspensions, which interact to form a continuous three-dimensional mesh structure extending throughout the volume. On this basis, the yield stress is related to the strength of the coherent network structure as the force needed to break down the structure, especially the network bonds or the breaking of the connecting bonds of the structural flow units. There have been many studies of the analysis of colloidal stability and solid particle surface properties under yield stress conditions. Many experimental methods have also been used to study the effect of particle concentration, size, particle size distribution, shape, surface activity, etc., on the yield stress values of various suspension systems [17,18,19,20,21]. The yield stress is the critical shear stress that causes irreversible plastic deformation and allows the fluid to flow in the pipe. The yield stress must be optimum to allow laminar transport of CAFB materials in the pipeline (velocity range 0.1 m/s to 1.5 m/s) without solids settling. The viscosity is the frictional resistance of two layers of concentrated fluid in the flow state. It is well known that the rheology of CAFB is influenced by various factors such as solid concentration, cement type and dosage, chemical additives and their dosage, particle size distribution, water chemistry and temperature. In addition, the microstructure of CAFB gradually changes during its transport due to the evolution of cement hydration products. CAFB is considered a non-Newtonian fluid because the shear stress at any point along the pipe cross-section during flow depends on the shear rate and time; therefore, the time-dependent rheology of CAFB is an issue of interest [22,23,24]. The water-reducing effect of plasticizers is mainly realized through the surface-active role of plasticizers. Plasticizers in the solid-liquid interface produce adsorption, reduce surface tension, and improve the surface-wetting state of cement particles, fly ash particles and gangue particles so that the mixture dispersion of thermodynamic instability is reduced to obtain relative stability [25,26,27,28,29]. Plasticizer produces directional adsorption on the surface between the particles so that the surface of the particles with the same charge, electrostatic repulsion, which destroys the flocculation-like structure between the particles, so that the particles are dispersed. For mines filled with CAFB material, the key to successful pipeline transport lies in the rheological properties of the filled slurry, which is ultimately determined by particle gradation. A reasonable material gradation should be selected to ensure uniform gradation, slow down the settling of aggregate particles to block the pipeline, and ensure that the slurry has good fluidity and stability with low overflow water over some time. To study the influence of the particle size gradation and hydration time of aeolian sand and coal gangue aggregate in different ratios on the rheological characteristics of the filling slurry, this paper elaborates on the intrinsic connection between the rheological indicators of the filling slurry and the particle gradation parameters through rheological tests, to provide a reference for the ratio test of aggregate, crushing and screening as well as the research of long-distance pumping and pressure conveying technology [8,15,16,30,31].

## 2. Rheological Model

### 2.1. Rheological Parameters Analysis

Yield stress and viscosity are the two basic parameters to characterize the rheology of a slurry. The yield stress can be divided into dynamic and static yield stress. The mass fraction of the slurry with yield stress is related to the content of fine particles. The smaller the particle size or, the higher the content, the lower the mass fraction of the slurry with yield stress. Viscosity reflects the size of the internal friction angle of the slurry flow itself, which is the macroscopic expression of the microscopic action of fluid molecules. The slurry viscosity is related to the particle size, distribution, mass fraction of solid particles, the energy exchange between solid and liquid particles and other factors related to the hydration reaction of cement and the generation rate of products of volcanic ash reaction [32].

### 2.2. The Rheological Model

The flow state of the material in different positions in the pipe can be roughly divided into “structural flow”, “laminar flow”, and “turbulent flow” according to the flow velocity, and the transmission characteristics are different from the two-phase flow movement law. When the mass fraction of the slurry reaches a certain level, the slurry becomes very viscous. The transmission characteristics change significantly along the pipeline and the slurry moves in a “plunger” shape. National and international researchers have confirmed that the Reynolds number of pipelines transporting high mass fraction (paste) slurry is much lower than the Reynolds number from laminar flow to turbulent flow, and the rheological model of high mass fraction slurry is suitable to use H-B model, whose rheological characteristic curve is shown in Figure 1:(1)$$\tau =\tau_{0}+\mu \gamma^{n}$$
where τ is the shear stress, Pa; γ is the shear rate, s^−1^, μ is the viscosity, Pa·s; $\tau_{0}$ is the initial yield stress, Pa; *n* is the fluidity index. When *n* = 1, $\tau_{0}$ = 0, it is a Newtonian body; when *n* = 1, $\tau_{0}>$ 0, it is the Bingham body; when *n* > 1, it is the swelling body; when *n* < 1, it is pseudoplastic body [33].

### 2.3. The Properties and Characteristics of Materials

The following details the characteristics of the materials used in this study, such as aeolian sand, coal gangue, cement, fly ash and chemical admixture.

#### 2.3.1. Aeolian Sand

The samples were collected near the Shiyaodian coal mine in Shenmu, Yulin, Shaanxi, China. The mineralogical analysis of the aeolian sand was carried out using an X-ray diffraction (XRD) spectrometer (Japan Rigaku SmartLab, Nagano, Japan), as shown in Figure 2. The particle size analysis was performed using a particle size analyzer (NKT5200-HF, Shandong Nexter Analytical Instruments Co., Ltd., Jinan, China), which showed that the particle size of the aeolian sand was mainly distributed between 100 μm and 300 μm in Figure 3. The chemical composition of the aeolian sand was analyzed by X-ray fluorescence (XRF) spectrometer, as shown in Table 1.

#### 2.3.2. Coal Gangue

The coal gangue samples were collected from Shenmu Shiyaodian coal mine, Yulin, Shaanxi, China. The mineralogical analysis of the coal gangue was carried out using an XRD spectrometer (Japan Rigaku Smartlab), as shown in Figure 4, and the coal gangue was crushed below 3 mm by mechanical processing. The particle size analysis was carried out using a particle size analyzer (NKT5200-HF), and the results showed that 50% of the coal gangue was less than 500 μm in size in Figure 5. and the chemical composition of the coal gangue was analyzed using an XRF spectrometer, as shown in Table 2.

#### 2.3.3. Fly Ash

Fly ash samples were collected from the Shenmu Guohua Power Plant in Yulin, Shaanxi, China. The mineralogical analysis of the fly ash was carried out using an XRD spectrometer (Japan Rigaku SmartLab), as shown in Figure 6, and particle size analysis was performed using a particle size analyzer (NKT5200-HF), which showed that the fly ash particle size mainly ranged from 0 μm to 300 μm in Figure 7. and the chemical composition of fly ash was analyzed using XRF spectroscopy, as shown in Table 3.

#### 2.3.4. Cement

The cement was ordinary Portland cement 425# (P.O. 42.5), the bulk density was 3100 kg/m^3^, and the initial and final setting times were 165 min and 231 min, respectively. The compressive strengths at 3d and 28d were 18.4 MPa and 46.4 MPa, respectively. The mineralogical analysis of the cement was carried out using an XRD spectrometer (Japan Rigaku Smartlab), as shown in Figure 8. The particle size analysis was performed using a particle size analyzer (NKT5200-HF), and the results showed that the cement particle size was mainly below 100 μm in Figure 9. The chemical composition of the cement was analyzed using an XRF spectrometer, as shown in Table 4.

#### 2.3.5. Chemical Admixture

Master Glenium 7500 (Master G), a chemical company Badische Andilin und Soda Fabrik (BASF) product, was used as the plasticizer in the CAFB mix. It is a full-range water-reducing admixture generally used for workability improvements and early strength achievement. It meets the ASTM requirements for high-range water-reducing admixtures [25,34].

### 2.4. Instructions for Rheometer and Vane Spindle

The rheological behavior of CAFB is measured using the RheolabQC rheometer (Anton paar RheolabQC, Germany) with a four-blade sensor [35]. The geometry of the vane sensor corresponds to a cylinder with a diameter of 22 mm and a height of 40 mm. The rheometer can be operated in a controlled shear stress or shear rate mode. The rheometer has a shear rate from 1 × 10^−3^ s^−1^ to 4 × 10^3^ s^−1^, shear stress from 0.5 Pa to 3 × 10^4^ Pa, depending on the measurement system, and viscosity from 1 Pa·s to 1 × 10^9^ Pa·s. The accuracy of the rheometer is 0.1%. The rheometer works on the principle of Searle. The vane spindle has an aspect ratio (height/diameter) 1.8. The geometry of the blade spindle effectively eliminates any wall slip effects, suppressing any significant interference with the sample that may result during the immersion of the blade into the sample before any measurement, thus allowing the paste to yield under the static conditions of the material itself. It will be very important that the suspension is thixotropic. In the wall slip phenomenon, an extremely thin film of low-concentration solid lubricant is formed in the material near the rotating surface of a concentric cylindrical or conical plate rheometer. As a result, the shear stress and viscosity in the film are much lower relative to the remaining dense, concentrated material. Therefore the true rheological behavior of the concentrated material may not be captured. Therefore, the vane is best suited for the thixotropic behavior of highly concentrated suspensions. It is important to note that the depth of the CAFB suspension and the diameter of the sample container (beaker or cup) should be at least twice the length and diameter of the vane to minimize the effects caused by rigid boundaries. For satisfactory measurements with the vane spindle, the following criteria are followed: H/D<3.5, $D_{T}/D>2.0$, $Z_{1}/D>1.0$, and $\;Z_{2}/D>0.5$ (Figure 7). The submerged depth of the vane is described by the distances $Z_{1}$ and $Z_{2}$, while H is the height of vane blades, as shown in Figure 10. Here, the diameters of the vane spindle and the sample container are calculated, respectively.

### 2.5. Specimen Preparation and Experimental Methods

The CAFB’s ingredients, i.e., solid concentration *C_wt.%_*, aeolian sand dosage *B_wt.%_*, coal gangue dosage *D_wt.%_*, fly ash dosage *E_wt.%_*, cement dosage *F_wt.%_*, plasticizer dosage *SP*(%) and water content *G_wt.%_* which are used in subsequent sections of this paper are calculated from the following equations (Equation (2a)–(2g)):(2a)$$C_{wt.\%}=\frac{100\times M_{dry\text{-}solid}}{M_{dry\text{-}solid}+M_{water}}$$
(2b)$$B_{wt.\%}=\frac{100\times M_{aeolian\text{-}sand}}{M_{dry\text{-}solid}+M_{water}}$$
(2c)$$D_{wt.\%}=\frac{100\times M_{coal\text{-}gangue}}{M_{dry\text{-}solid}+M_{water}}$$
(2d)$$E_{wt.\%}=\frac{100\times M_{fly\text{-}ash}}{M_{dry\text{-}solid}+M_{water}}$$
(2e)$$F_{wt.\%}=\frac{100\times M_{cement}}{M_{dry\text{-}solid}+M_{water}}$$
(2f)$$SP(\%)=\frac{100\times M_{sp}}{M_{cement}}$$
(2g)$$G_{wt.\%}=\frac{100\times M_{water}}{M_{dry\text{-}solid}+M_{water}}$$
where *M_water_* is the mass of water in the paste fill; *M_dry-solid_* is the mass of aeolian sand, coal sand, Portland cement, and fly ash. *M_aeolian-sand_* is the mass of aeolian sand, *M_cement_* is the mass of Portland cement, *M_coal-gangue_* is the mass of coal gangue, *M_fly-ash_* is the mass of fly ash, and *M*_sp_ is the mass of plasticizer. The amount of aeolian sand and coal gangue are varied to prepare different mix compositions, as shown in Table 5.

#### 2.5.1. Vane Test to Determine Yield Stress and Viscosity

The following outlines the procedural steps for developing a slurry material for rheological evaluation.

**Step 1. Initial mixing:** The air-dried coal gangue is thoroughly mixed with a certain proportion of aeolian sand, cement, and fly ash using a spatula in a mixing container. Then specified amount of tap water followed by a certain proportion of SP is added and homogenized thoroughly using the electric blender at high shear (+1000 rpm) for 5 min.

**Step 2. Final mixing:** The slurry was then loaded into a 500 mL beaker, and finally, the container loaded with the slurry was assembled with the rheometer for subsequent rheological tests, with a four-blade rotor gradually entering the middle of the beaker and inserted with the slurry sample undisturbed for 30 s to allow the mixed slurry to reach its structural balance and at least partially recover its initial structure and strain state.

**Step 3. Determination of yield stress and viscosity**: Rheological curves describing the correlation between shear stress and shear rate were obtained experimentally for each sample. The rheological parameters describing the rheological properties of the slurry included yield stress and viscosity. The yield stress is the stress value at which the material starts to yield to a state transition, and the shear rate of the rheological curve is from 0.01 s^−1^ to 150 s^−1^ with an interval of 0.05 s^−1^. The Herschel-Bulkley rheological model was used to fit the measured rheological curves. In the fitted curves, the intersection of the shear stress curve with the longitudinal coordinate (longitudinal intercept) is the yield stress, the viscosity is the slope of the fitted curve, and the viscosity is the ratio of shear stress to shear rate.

**Step 4.** The above process was repeated for samples from the same slurry batch to obtain different parameters, i.e., 3 min, 30 min, and 60 min after sample preparation. The hydration age was chosen to cover various transport times encountered during the slurry backfill tests. The shear test was performed using a controlled shear rate by placing the rotor in a 500 mL beaker for rheological testing, rotating it at a variable shear rate, dispensing the slurry several times and taking the average value for multiple measurements to eliminate experimental errors, and recording the corresponding shear stress and viscosity in real-time. The test is repeated three times for each slurry sample to ensure reproducible results.

#### 2.5.2. Differential Thermogravimetric Analysis Test

To gain a more in-depth understanding of the effects of different factors on the cement hydration process in CAFB, microstructure analyses, including thermal analyses thermogravimetry (TG) and differential thermogravimetry (DTG), were conducted on the prepared cement fly ash paste (CFP) samples of the CAFB. The CFPs were prepared at a water/binder ratio of 1, and cured at room temperature for 1 h and 2 h, respectively. Before testing, the specimens were dried in an oven-dried at 45 °C until a constant mass was attained.

The thermal analyses were carried out using a simultaneous thermogravimetric analyzer and differential scanning calorimeter (SDT) from TA Discovery TGA 550. The dried samples were placed on a plate and heated under an inert nitrogen atmosphere at 10 °C/min from room temperature to 800 °C.

#### 2.5.3. Zeta Potential Test

The zeta potentials of CAFB samples treated with a different plasticizer dosage were measured using the Malvern Zetasizer Nano series. The device measures the electrophoretic mobility of suspended particles based on phase analysis light scattering (PALS). The zeta potential of the particles is evaluated using the Henry equation.

#### 2.5.4. Monitoring through Electrical Conductivity

To gain additional insight into the reaction mechanisms responsible for the rheological behavior of the CAFB with varying compositions, a 5TE sensor was used to monitor the electrical conductivity (EC), while an MPS-6 sensor was used for suction. Each sensor was placed in the middle of a cylindrical plastic container of 10 cm diameter by 20 cm height filled with the CAFB, and readings were recorded from the time of cast until 2 h.

#### 2.5.5. Scanning Electron Microscopy Morphological Analysis

The parts of the dried CAFB specimens were cut into small rectangular specimens (10 mm × 10 mm × 10 mm) with a naturally formed surface. The specimens were placed in a holder with its natural surface facing upwards. The base of the specimens was attached to the holder using conductive adhesive or glue. Finally, micrographs of the prepared SEM specimens were captured using a ZEISS Gemini SEM 300 scanning electron microscope with magnifications between 100× and 5000× (Carl Zeiss AG, Berlin, Germany).

## 3. Results and Discussion

CAFB material is added with different proportions of samples of aeolian sand and coal gangue, along with the growth of hydration time, the rheological parameters yield stress and viscosity of the slurry are measured, and the influence of particle gradation of the slurry and the rheological parameters of the slurry are counted by linear regression method to explore the best combination of material gradation, and reduce the resistance loss of the material in the process of pipeline conveying, and establish the corresponding basis.

### 3.1. Fitting and Analysis of Rheological Parameters

The linear regression analysis of the experimental results was performed to obtain the indicators of the rheological properties of the slurry under different conditions by Equation (1), as shown in Table 6. From Table 6, it can be concluded that with the extension of the resting time of the slurry, the values of the parameters reflecting the rheological characteristics of the slurry change inconsistently, the initial shear stress of 1#, 2#, 4# and 5# of slurry all increase first and then decrease, and the viscosity of 1#, 3#, 4# and 5# of slurry all increase first and then decrease; with the same resting time, the rheological characteristics of the slurry composed of different materials are different. In the resting time of 30 min, the initial shear stress of 4# and 5# slurry is larger than the other three groups of slurry. The initial shear stress of 4# and 5# slurry is greater than the other three groups because the particle gradation of 4# and 5# slurry is similar, and the volume fraction of fine particles contained in the unit volume is greater. The rheological model of the slurry is changing as time grows, the hydration reaction inside the slurry continues, and the number of flocculent network gel products with resistance to mechanical damage is increasing, and the strength is increasing. Comparing the rheological indexes of the characterized slurry in Table 6, it can be concluded that 1#, 2#, 3#, 4# and 5# of slurry are in swelling body after 3 min of rest, and their rheological indexes are greater than 1. 3# and 4# of slurry are in Bingham body after 30 min of rest, and their rheological indexes are close to 1. 1# and 5# of slurry are in the pseudoplastic body after 60 min of rest, and their rheological indexes are less than 1 [9,11].

### 3.2. The Influence of the Amount of Coal Gangue Added on the Rheological Properties of the Slurry

From Figure 11, Figure 12 and Figure 13, the relationship between shear stress and shear rate is different from different grades of slurry at different shear rates. The rheological characteristic curves of 3#, 4# and 5 # of slurry are the same, and the rheological characteristic curves of 1# and 2# of slurry are the same with similar basic laws. With a shear rate from 10 to 60, the rheological curves of slurry are pseudoplastic body; with a shear rate from 60 to 100, the rheological curves of slurry are Bingham body; with a shear rate from 100 to 150, the rheological curves of slurry are pseudoplastic body with certain yield stress; with the increasing shear rate, the shear stress is also increasing, but the particle gradation of the slurry is not consistent, resulting in the increase of the shear stress of the slurry from 1# to 5# is not the same degree. The initial shear stress is not the same with the increase of the mass fraction of the unit volume in the slurry. The initial shear stress of the slurry is also increasing with the increase of the unit volume mass fraction in the slurry, which is consistent with the particle size gradation. From 1# to 5# slurry, the viscosity is in the unstable stage, transition stage, and stable stage, and the value is decreasing, and the stable stage is below 10 Pa·s. From Table 7, the median particle size of 1# to 5# slurry is 138.018 μm, 151.538 μm, 165.059 μm, 178.579 μm and 192.149 μm respectively. The distribution of the particle size of coal gangue can be seen that 50% of the coal gangue is below 500 μm. The particle size of the aeolian sand is mainly between 100 and 300 μm. From the slurry Proportional distribution, it can be seen that from 1# to 5# slurry, the proportion of aeolian sand is in decline, but the proportion of coal gangue is on the rise. The mass fraction of unit volume is increasing, so the median particle size of 1# slurry to 5# slurry is also increasing, and the D_10_ particle size value is becoming smaller. The D_90_ particle size value is in becoming larger, which indicates that the particle size distribution of the material is more uniform, but, with the increase of coal gangue, the initial shear stress of the slurry is getting bigger and bigger, the reasonable particle gradation should not only ensure that the slurry has good rheology but also make the yield stress of slurry to be in a reasonable range, so that the pipeline’s conveying resistance runs in a reasonable range and reduce the energy loss [26,36].

### 3.3. Correlation between Rheological Parameters and Particles Size

If there is a certain correlation between the rheological parameter X_1_ and the particle gradation parameter X_2_ for different slurries, the correlation is expressed by the following equation.
(3)$$x_{1}=a+bx_{2}$$

From the principle of least squares, the correlation coefficient r between the parameters X_1_ and X_2_ can be derived. In general, when r < 0.4, there is no or weak correlation between the variables; when 0.4 < r < 0.6, there is a moderate correlation between the variables; when 0.6 < r < 0.8, there is a strong correlation (significant correlation) between the variables; and when 0.8 < r < 1.0, there is a very strong correlation between the variables.

The linear regression results between the different parameters were obtained by least squares calculation of the parameters in Table 6 and Table 7 according to the correlation equation. The correlations between the viscosity and yield stress of the slurry at 3, 30 and 60 min of standing are shown in Table 6, and the particle grading parameters are shown in Table 7, respectively. From Table 8, it can be concluded that the correlation coefficients of yield stress and D_10_, D_30_, D_50_, D_60_ and D_90_ of particle size after 3 min of standing are above 0.98, which indicates that there is a strong correlation between them because the mass fraction of the slurry with yield stress is related to the content of fine particles, and the smaller the particle size or, the higher the content, the lower the mass fraction of the slurry with yield stress. with the growth of coal gangue particle content, 1#, 2#, 3#, 4# and 5# of mixed slurry fine particles are increasing, and the proportion of coarse particles is also increasing, which shows that there is a strong correlation between the yield stress of the slurry and the slurry particle size. with the slurry in the rest 3 min, the correlation coefficient of viscosity and D_10_, D_30_, D_50_, D_60_ and D_90_ of particle size is above 0.75. This shows that there is a strong correlation between the two, which indicates that there is also a strong correlation between the particle gradation and the viscosity of the slurry.

Table 9, can be derived from the slurry after 30 min resting. The correlation coefficients of yield stress and D_10_, D_30_, D_50_, D_60_ and D_90_ of particle size is above 0.86 or more, which indicates that there is a strong correlation between the two, because the mass fraction of the slurry with yield stress is related to the content of fine particles, the smaller the particle size or the higher the content, the slurry mass fraction of yield stress is lower. with the growth of coal gangue particle content, 1#, 2#, 3#, 4# and 5# of mixed slurry fine particles are increasing, and the proportion of coarse particles is also increasing, which shows that there is a strong correlation between the yield stress of the slurry and the slurry particle size. With the slurry in the resting 30 min, the correlation coefficient between viscosity and D_10_, D_30_, D_50_, D_60_ and D_90_ of particle size reached 0.20, indicating no correlation between them.

Table 10 can be derived from the slurry in the rest 60 min. The correlation coefficient of initial yield stress and D_10_, D_30_, D_50_, D_60_ and D_90_ of particle size are about 0.93 or more, indicating that there is a strong correlation between the two because the mass fraction of the slurry with yield stress is related to the content of fine particles, the smaller the particle size or, the higher the content, the slurry mass fraction of yield stress is lower. with the growth of coal gangue particle content, 1#, 2#, 3#, 4# and 5# of mixed slurry, fine particles are increasing, and the proportion of coarse particles is also increasing, which shows that there is a strong correlation between the yield stress of the slurry and the slurry particle size. with the slurry in the resting 60 min, the correlation coefficient of viscosity and D_10_, D_30_, D_50_, D_60_ and D_90_ of particle size are 0.39 or more, indicating a weak correlation between the two.

From Table 8, Table 9 and Table 10, it can be concluded that the initial shear stress of the slurry follows a very strong correlation (3 min–30 min–60 min) with the slurry resting time, and on the contrary, the viscosity of the slurry follows a strong correlation (3 min)—no correlation (30 min)—weak correlation (60 min). Under the same conditions, the particle sizes of D_10_, D_30_, D_50_, D_60_ and D_90_ greatly influence the relationship with the initial shear stress of the slurry. At 3 min, the viscosity of each particle size with the slurry has a greater influence.

### 3.4. Time-Dependent Evolution of the Rheological Properties of CAFB with the Ratio of Aeolian Sand/Coal Gangue

The variation of the yield stress between 0 h and 1 h for different types of slurry 1#–5# is shown in Figure 14. The average density of 1#–5# slurry is 2.2 g/cm^3^.

It shows that the curing time has a great influence on the yield stress value of 1#–5# slurry in Figure 14, in which the yield stress value of 1# slurry increases from 415.16 Pa at 0 h to 611.65 Pa at 0.5 h; this is mainly because the hydration reaction of cement in the slurry increases with time and the hydration products keep increasing, which causes the yield stress value of the slurry to become largely. It increases the frictional resistance between the fly ash particles and gangue particles of CAFB, which enlargements the yield stress [18]. Furthermore, more hydration products will significantly increase the cohesion among cement particles, fly ash particles and gangue particles, increasing yield stress [36,37]. During the period of 0.5–1 h, the yield stress values of 1#, 4# and 5# slurry were significantly reduced from 611.65 Pa, 779.76 Pa and 919.08 Pa to 410.89 Pa, 725.24 Pa and 705.04 Pa. This is because the increase of cement hydration products C-S-H, CH wrapped in aeolian sand, and coal gangue surface fill the pores therein, significantly reducing the yield stress of CAFB.

The TG/DTG analysis results experimentally support the increase in hydration products due to the longer maintenance period in Figure 15. It shows the TG/DTG analysis results of the slurry of CAFB samples for 1 h and 2 h. The DTG curve shows two main internal heat peaks related to the rapid weight loss and the main phase transition. The first internal heat peak appeared at 102.92 °C, and the weight loss occurred between 114 °C and 116 °C, and the weight change was 0.430% within 1 h. However, the first internal heat peak appeared at 51.36 °C, where the weight loss occurred between 48.13 °C and 81.91 °C, and the weight change was 0.997% within 2 h, which was the reason for the dehydration of hydration products, such as C-S-H, carbo aluminates, ettringite, and gypsum [38]. Previous studies have also concluded that the decomposition of gypsum and ettringite and the dehydration of some carbo aluminate hydrates occur in the temperature range of 60–200 °C. Two internal heat peaks were found at 637.09 °C, and the weight loss was found between 585.76 °C and 658.12 °C, and the weight change was 6.552% within 1 h. However, two internal heat peaks were found at 620.26 °C, and the weight loss was between 575.44 °C and 639 °C. Comparing the TG/DTG diagrams of the slurry cured for 1 h and 2 h, the latter has more weight loss or higher peaks in the 60–200 °C temperature range. It indicates that the CAFB maintained for a long time will have more cement hydration products (C-S-H, ettringite, etc.) than the samples maintained for a shorter time. It is obtained that the maintenance period for the cement hydration process is consistent with the previous conclusion [39].

### 3.5. Time-Dependent Evolution of the Rheological Properties of CAFB with Plasticizer

Viscosity and yield stress were used to study the rheology of fresh CAFB made with different plasticiser ratios. It shows the development of yield stress in the first 1 h of mixing in Figure 16. Likewise, adding the plasticizer led to a proportionate reduction in the yield stress of CAFB [1,20]. In the maintenance period of 1 h, the yield stress decreased more obviously. The yield stress of 0.05% admixture content decreased from 652 Pa to 234 Pa, and the yield stress of 0.1% admixture content decreased to 215 Pa. The increased fluidity observed from the 0.05% and 0.1% admixture content may be related to the dispersion effect of the plasticizer. At the early stages of preparation, the viscosity of CAFB without addition was a lot higher than that with the addition of 0.05% and 0.1%, as shown in Figure 17. With moderate additions of 0.05%, the viscosity decreased from 0.148 Pa·s to about 0.04 Pa·s at 0 h, and this tendency continued for the rest of the maintenance period. It can be seen that the Zeta potential of CAFB with and without the mixture supports this point in Figure 18. The zeta potential of CAFB containing 0.05% mixture was about −10 mV, while the untreated sample was −3.5 mV. This indicates that the repulsion between particles is much higher in the presence of plasticizers [37]. Another explanation for the change in CAFB fluidity caused by adding plasticizers is its effect on cement hydration. Due to the increase of repulsive force among cement particles, fly ash particles and gangue particles, less hydration by-products are produced, and the viscosity of CAFB decreases with the decrease of solid content [40,41,42,43]. In addition, the EC monitoring results support the conclusion that the plasticizer postpones the hydration reaction in Figure 19. The EC of CAFB containing 0.1% plasticizer was lower than that containing 0% CAFB. Therefore, the mobility of ions is higher in CAFB without additives [44,45].

It can also be seen that the yield stress of CAFB samples increases smoothly with the maintenance period owing to the hydration process of cement and the surface absorption of poorly crystalline C-S-H gels in Figure 16. Once the cement is in contact with water, the chemical reaction usually begins [22]. In addition, it can be seen by electron microscope scan that CAFB with a maintenance age of 3 d can see C-S-H, CH, and ettringite filling around the aeolian sand and coal gangue, which affects the fluidity of CAFB slurry, as shown in Figure 20 [20,41,42,46,47,48].

## 4. Conclusions

The time-dependent rheological properties of fresh CAFB treated with the Master Glenium plasticizer and cured under different ratio of aeolian sand/coal gangue were studied in this paper. Plasticizer dosages of 0, 0.05, and 0.1% of the total CAFB weight were used. Additional tests, such as thermal analysis and zeta potential analysis, were conducted to understand the reasons behind the nature of the results observed. The major conclusions based on the results obtained are summarized below.

The 3#, 4# and 5# slurry flow index change patterns are the same. The index n, which characterizes the flow property as greater than 1 after 3 min resting, belongs to the swelling body; *n* is close to 1 after 30 min resting, belongs to Bingham body; *n* is less than 1 after 60 min resting, belongs to the pseudoplastic body; 1# and 2# slurry in 3–30 min flow index is greater than 1, belongs to the swelling body, After 60 min of resting, the flow pattern of slurry changes *n* is less than 1, which belongs to the pseudoplastic body.With the increase of shear rate and shear time, the viscosity first gradually decreases and then stabilizes, i.e., the rheological properties of the slurry have the characteristic of “shear thinning”; the rheological properties of the slurry process is the comprehensive embodiment of a variety of model composite properties, with the increase of shear rate, the rheological curve of the slurry shows an upward convex shape, showing a pseudoplastic body-Bingham body-Pseudoplastic body (swelling body).According to 2# and 3# mixed material, it will be grading configuration aeolian sand, coal gangue, and fly ash filling slurry to ensure long-distance pipeline conveying process slurry flow stability, to ensure smooth pipeline transport.Curing time (0–0.5 h) results in higher yield stress of the CAFB. A longer curing time is associated with a greater degree of cement hydration products. Curing time (0.5–1 h) results in lower yield stress of the CAFB. It is because more hydration products are associated with a decrease in the aeolian sand inter-particle frictional resistance of the CAFB.The addition of plasticizer to the CAFB significantly reduces the yield stress and viscosity of CAFB. Adding 0.05% of the admixture results in over 65% reduction in yield stress at the time of preparation and after 1 h. A similar reduction was observed in the improvement of viscosity. The marginal reduction upon increasing the admixture to 0.1% is much less, indicating that an optimum percentage is around 0.05.

## Figures and Tables

**Figure 1 materials-16-05295-f001:**
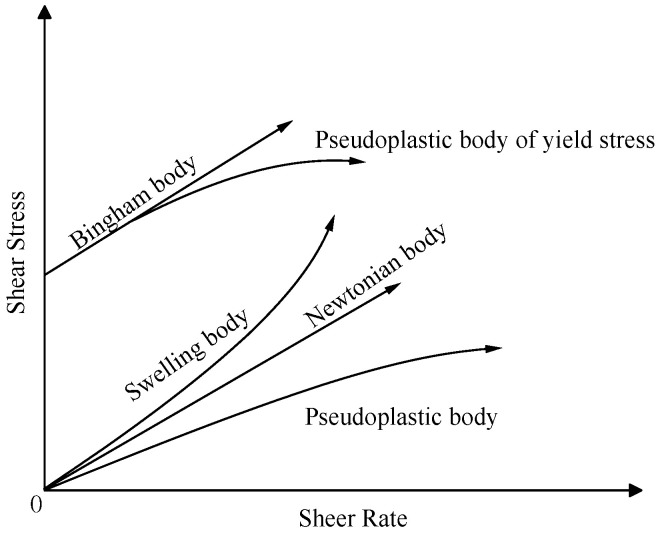
Slurry rheological characteristic curve.

**Figure 2 materials-16-05295-f002:**
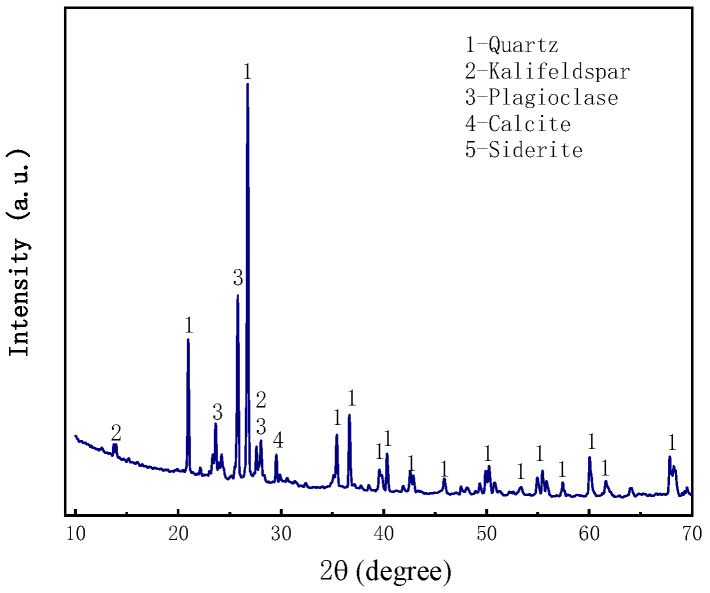
X-ray diffractogram of aeolian sand.

**Figure 3 materials-16-05295-f003:**
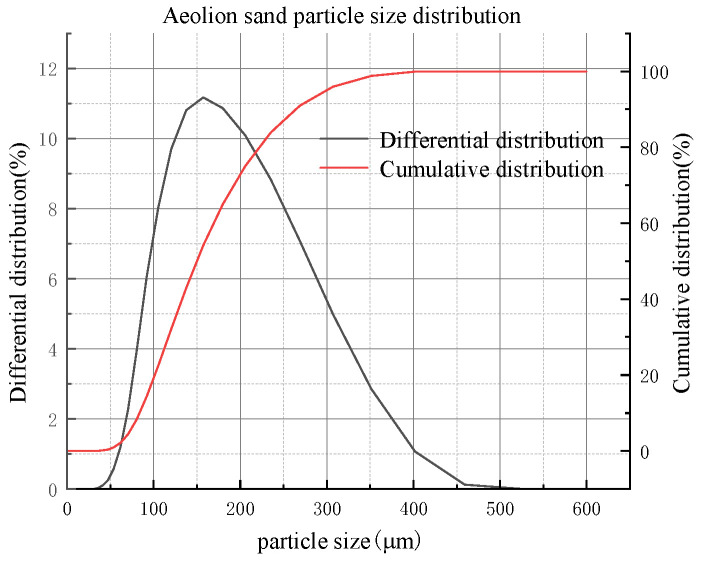
Particle size distribution of aeolian sand.

**Figure 4 materials-16-05295-f004:**
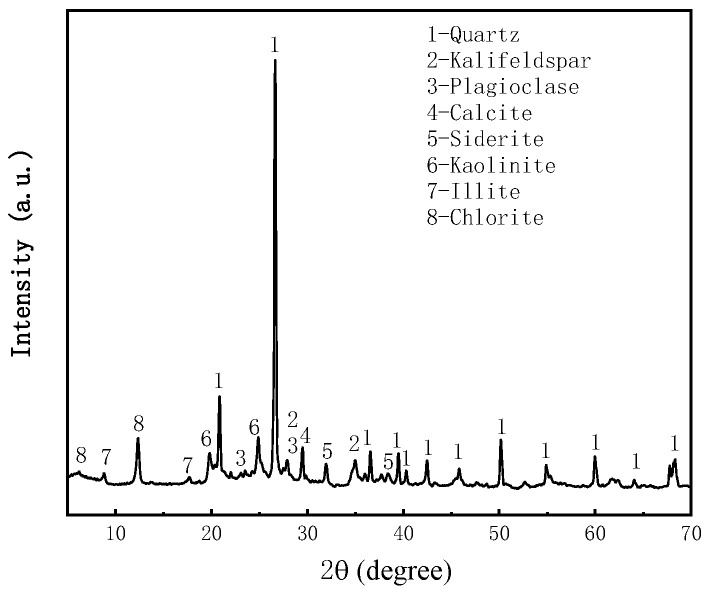
X-ray diffractogram of coal gangue.

**Figure 5 materials-16-05295-f005:**
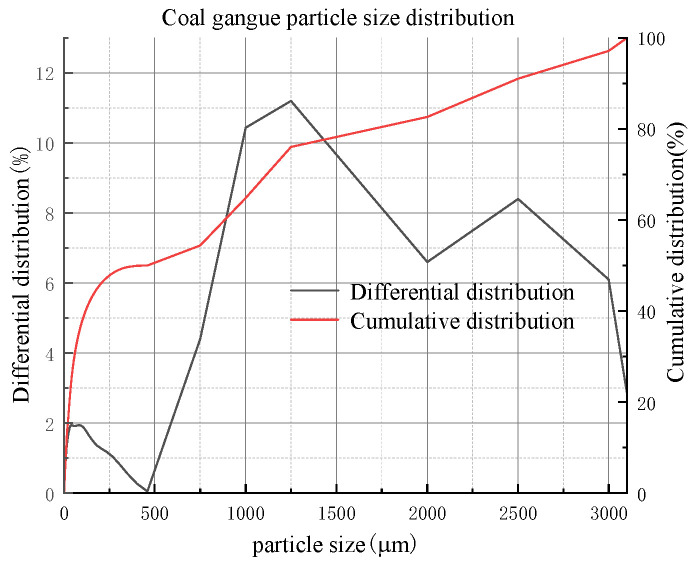
Particle size distribution of coal gangue.

**Figure 6 materials-16-05295-f006:**
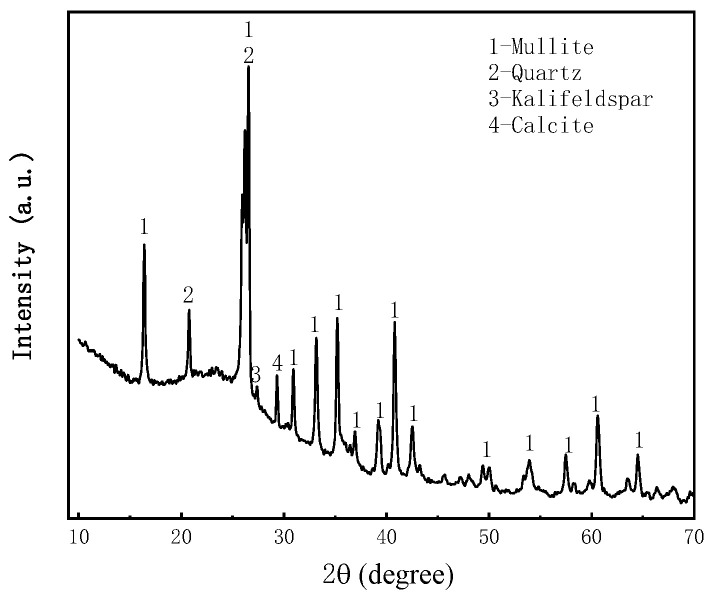
X-ray diffractogram of fly ash.

**Figure 7 materials-16-05295-f007:**
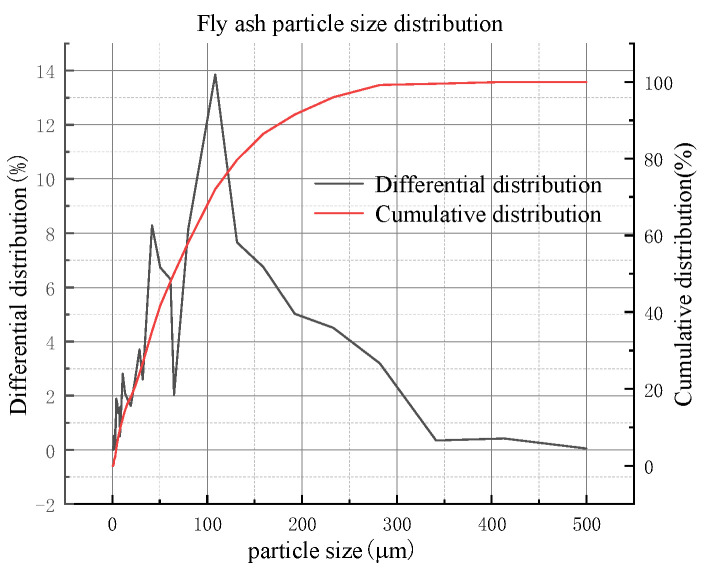
Particle size distribution of fly ash.

**Figure 8 materials-16-05295-f008:**
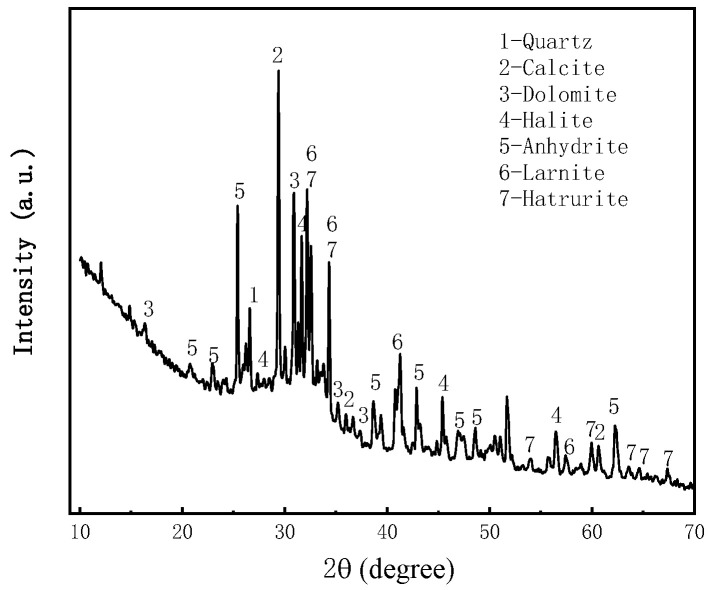
X-ray diffractogram of cement.

**Figure 9 materials-16-05295-f009:**
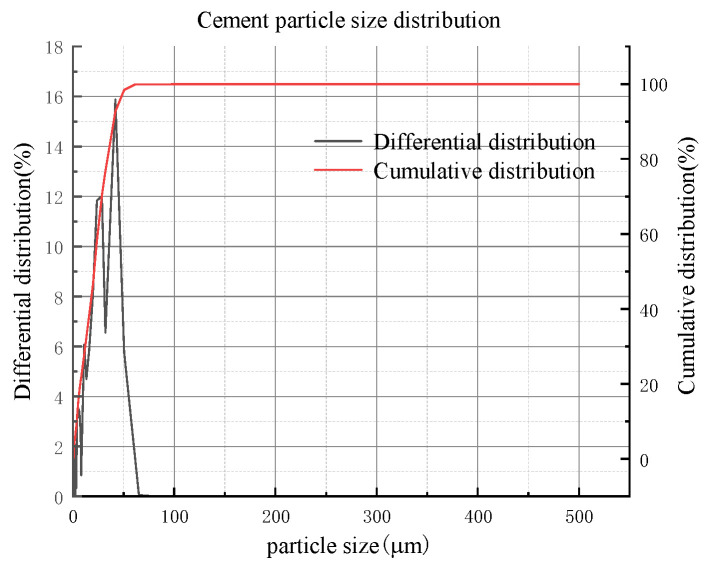
Particle size distribution of cement.

**Figure 10 materials-16-05295-f010:**
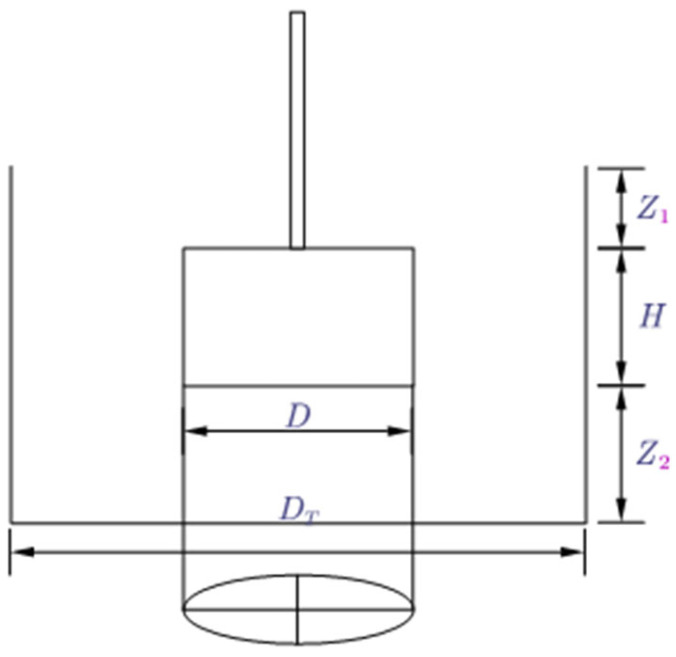
Dimensions of the four-bladed vane and sample container.

**Figure 11 materials-16-05295-f011:**
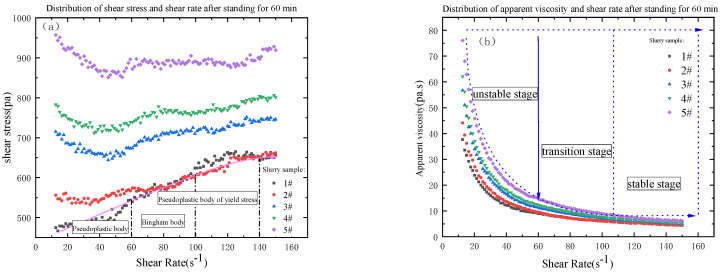
Distribution diagram of shear stress (**a**), viscosity (**b**) and shear rate after standing for 60 min.

**Figure 12 materials-16-05295-f012:**
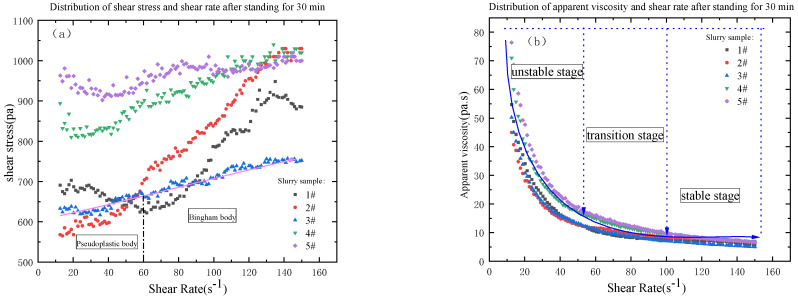
Distribution diagram of shear stress (**a**), viscosity (**b**) and shear rate after standing for 30 min.

**Figure 13 materials-16-05295-f013:**
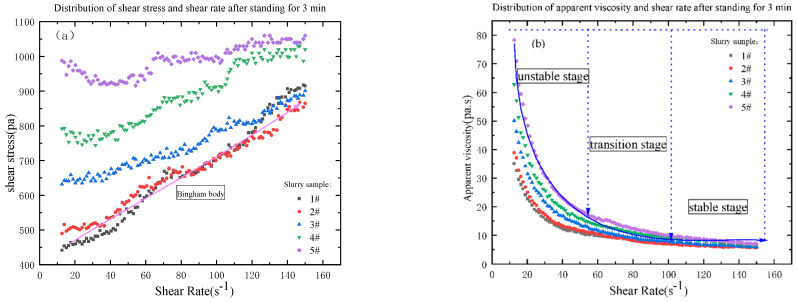
Distribution diagram of shear stress (**a**), viscosity (**b**) and shear rate after standing for 3 min.

**Figure 14 materials-16-05295-f014:**
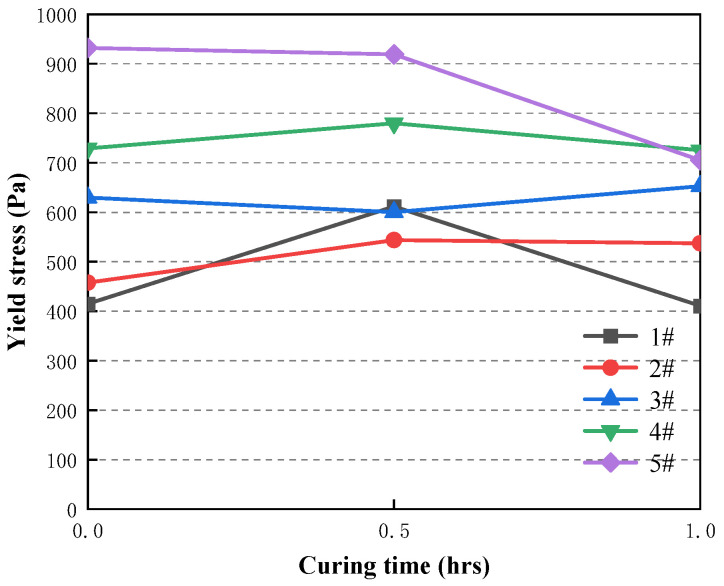
Time-dependent evolution of the yield stress of fresh CAFB.

**Figure 15 materials-16-05295-f015:**
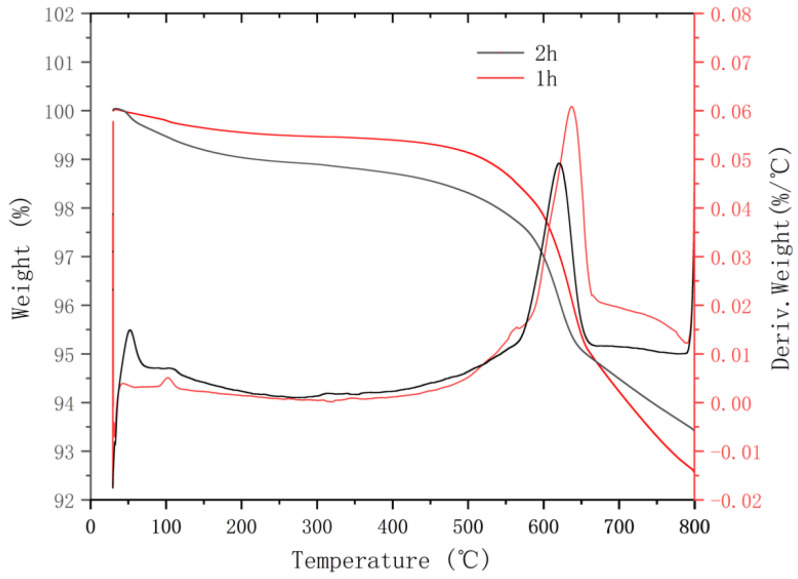
Plotted TG/DTG for cement paste of CAFB cured for 1 h and 2 h.

**Figure 16 materials-16-05295-f016:**
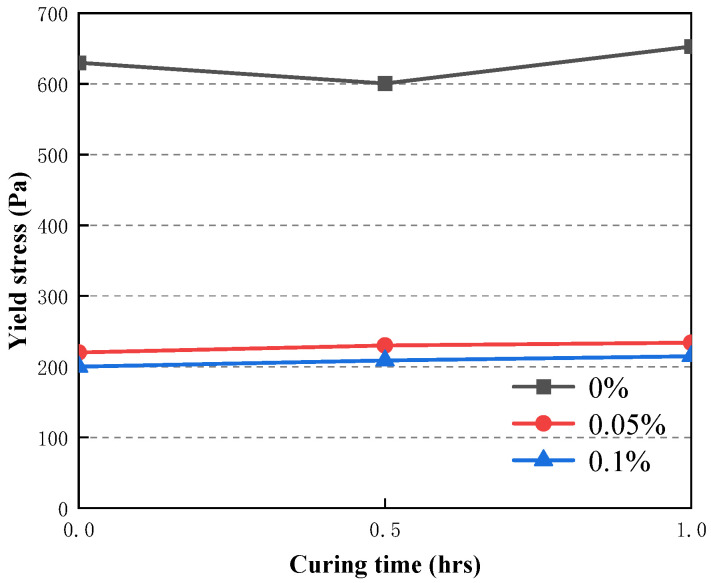
Time-dependent evolution of yield stress of CAFB containing different dosages of plasticizer.

**Figure 17 materials-16-05295-f017:**
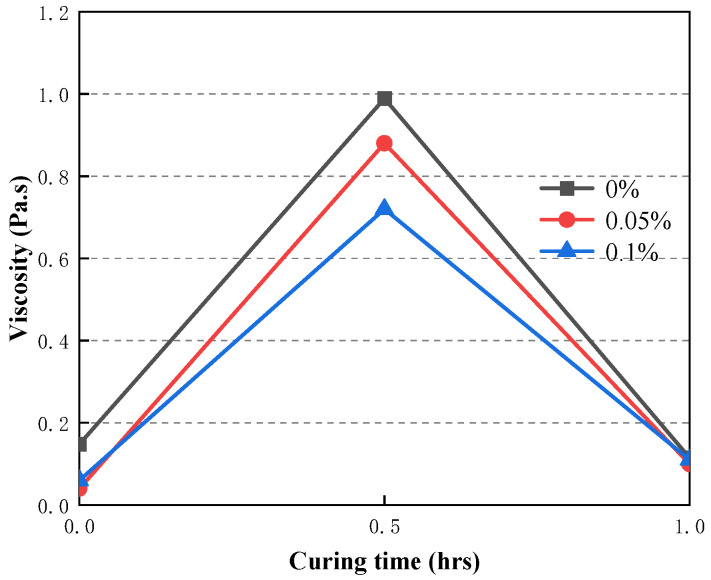
Time-dependent evolution of viscosity of CAFB containing different dosages of plasticizer.

**Figure 18 materials-16-05295-f018:**
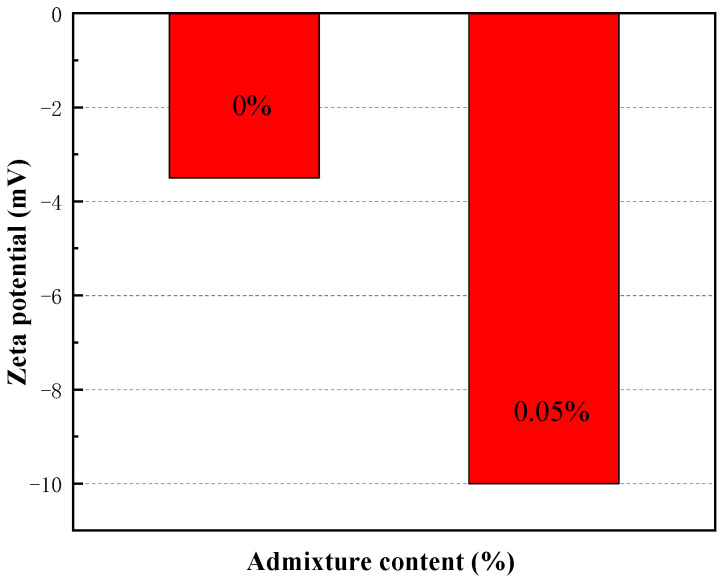
Effect of the plasticizer on the zeta potential of fresh CAFB made with different admixture content.

**Figure 19 materials-16-05295-f019:**
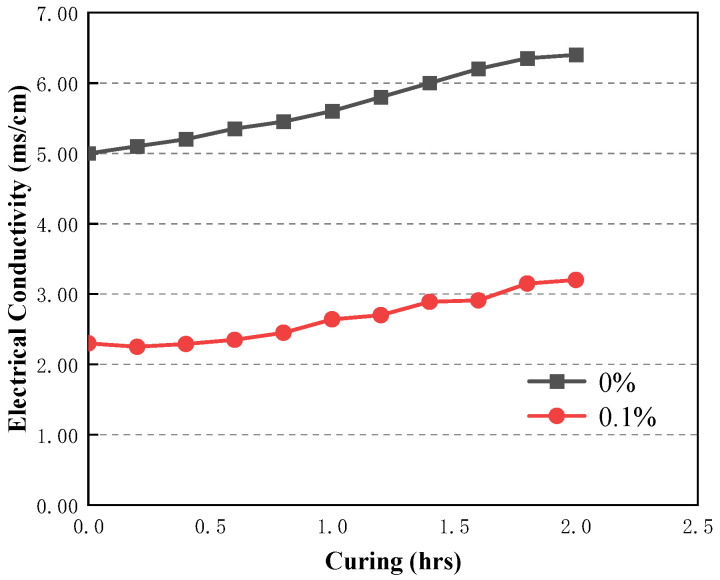
Development of electrical conductivity in CAFB containing different dosages of plasticizer.

**Figure 20 materials-16-05295-f020:**
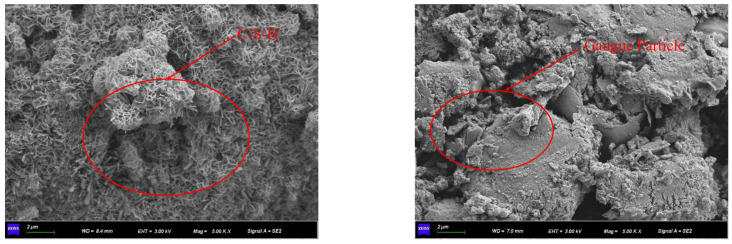
ESEM micrographs of 3d CAFB.

**Table 1 materials-16-05295-t001:** Results of semi-quantitative analysis of aeolian sand energy spectrum.

Chemical Composition	Al_2_O_3_	SiO_2_	Na_2_O	CaO	K_2_O_4_	Fe_2_O_3_	Other	Total
Mass percentage (%)	9.78	68.94	2.34	6.65	2.13	2.24	7.92	100

**Table 2 materials-16-05295-t002:** Results of semi-quantitative analysis coal gangue energy spectrum.

Chemical Composition	Al_2_O_3_	SiO_2_	P_2_O_5_	K_2_O	CaO	TiO_2_	Fe_2_O_3_	Total
Mass percentage (%)	22.23	62.06	0.65	2.95	3.11	0.99	8.01	100

**Table 3 materials-16-05295-t003:** Results of semi-quantitative analysis fly ash energy spectrum.

Chemical Composition	Al_2_O_3_	SiO_2_	S	K_2_O	CaO	TiO_2_	Fe_2_O_3_	Total
Mass percentage (%)	31.89	56.89	0.66	1.39	1.84	1.95	5.38	100

**Table 4 materials-16-05295-t004:** Results of semi-quantitative analysis of cement energy spectrum.

Chemical Composition	CaO	SiO_2_	Al_2_O_3_	Fe_2_O_3_	MgO	Other	Total
Mass percentage (%)	65.08	22.36	5.53	3.46	1.27	2.30	100

Note: Carbon and water are not analyzed in the semi-quantitative analysis results of the energy spectrum. If carbon and water are analyzed, the analysis results of other elements should be reduced accordingly.

**Table 5 materials-16-05295-t005:** CAFB mixture proportions.

CAFB Code	$B_{wt.\%}$	$D_{wt.\%}$	$E_{wt.\%}$	$F_{wt.\%}$	SP%	$G_{wt.\%}$	$C_{wt.\%}$
1#	46	5	17.5	9	0	22.5	77.5
2#	43	8	17.5	9	0	22.5	77.5
3#	40	11	17.5	9	0	22.5	77.5
4#	37	14	17.5	9	0	22.5	77.5
5#	34	17	17.5	9	0	22.5	77.5
6#	40	11	17.5	9	0.05	22.5	77.5
7#	40	11	17.5	9	0.1	22.5	77.5

**Table 6 materials-16-05295-t006:** Rheological parameters of filling slurry under different conditions.

Time/min	Rheology Index	1#	2#	3#	4#	5#
60	*μ* (Pa·s)	0.052	0.038	0.115	0.017	0.0104
	*τ*_0_/Pa	410.891	537.44	652.84	725.24	705.04
	*n*	0.788	1.636	1.359	1.682	0.121
	*R* ^2^	0.954	0.972	0.852	0.831	0.409
30	*μ* (Pa·s)	1.12	0.385	0.989	1.781	0.384
	*τ*_0_/Pa	611.650	543.827	600.575	779.76	919.08
	*n*	3.048	1.452	1.024	0.997	1.085
	*R* ^2^	0.956	0.989	0.964	0.953	0.626
3	*μ* (Pa·s)	1.166	1.763	0.148	0.563	0.041
	*τ*_0_/Pa	415.16	457.91	629.84	728.86	931.98
	*n*	1.210	1.080	1.496	1.269	1.621
	*R* ^2^	0.982	0.978	0.981	0.961	0.802

**Table 7 materials-16-05295-t007:** Characteristic index of particle size gradation of CAFB.

CAFB Code	D_10_/μm	D_30_/μm	D_60_/μm	D_90_/μm	D_50_/μm
1#	53.771	80.586	179.465	361.146	138.018
2#	51.263	78.586	207.453	449.321	151.538
3#	48.755	76.587	235.442	533.496	165.059
4#	46.247	74.587	263.43	617.671	178.579
5#	43.74	72.594	291.507	702.089	192.149
I 6#	48.755	76.587	235.442	533.496	165.059
7#	48.755	76.587	235.442	533.496	165.059

Note: D_10_, D_30_, D_60_, and D_90_ are the corresponding particle sizes when the cumulative particle size content reaches 10%, 30%, 60%, and 90%, respectively.

**Table 8 materials-16-05295-t008:** The correlation between the rheological parameters of the slurry and the particle size gradation after standing for 3 min.

Correlation Model	a	b	$\left|r\right|$	$\left|r^{2}\right|$
$\mu =a+bd_{10}$	−5.971	0.138	0.753	0.423
$\mu =a+bd_{30}$	−12.486	0.173	0.753	0.423
$\mu =a+bd_{50}$	4.945	−0.026	0.753	0.423
$\mu =a+bd_{60}$	3.637	−0.012	0.753	0.423
$\mu =a+bd_{90}$	2.890	−0.004	0.750	0.417
$\tau =a+bd_{10}$	3169.028	−52.021	0.981	0.951
$\tau =a+bd_{30}$	5632.278	−65.278	0.981	0.951
$\tau =a+bd_{50}$	−959.012	9.643	0.981	0.951
$\tau =a+bd_{60}$	−464.177	4.659	0.981	0.951
$\tau =a+bd_{90}$	−183.410	1.532	0.980	0.947

**Table 9 materials-16-05295-t009:** The correlation between the rheological parameters of the slurry and the particle size gradation after standing for 30 min.

Correlation Model	a	b	$\left|r\right|$	$\left|r^{2}\right|$
$\mu =a+bd_{10}$	0.784	0.003	0.021	0.333
$\mu =a+bd_{30}$	0.647	0.004	0.020	0.333
$\mu =a+bd_{50}$	1.027	−0.0005	0.021	0.333
$\mu =a+bd_{60}$	0.997	−0.0002	0.021	0.333
$\mu =a+bd_{90}$	0.985	−0.0001	0.023	0.333
$\tau =a+bd_{10}$	2344.996	−33.924	0.868	0.672
$\tau =a+bd_{30}$	3951.053	−42.566	0.868	0.671
$\tau =a+bd_{50}$	−347.240	6.290	0.868	0.672
$\tau =a+bd_{60}$	−24.474	3.038	0.868	0.672
$\tau =a+bd_{90}$	160.503	0.996	0.864	0.662

**Table 10 materials-16-05295-t010:** The correlation between the rheological parameters of the slurry and the particle size gradation after standing for 60 min.

Correlation Model	a	b	$\left|r\right|$	$\left|r^{2}\right|$
$\mu =a+bd_{10}$	−0.156	0.004	0.395	0.126
$\mu =a+bd_{30}$	−0.353	0.005	0.395	0.126
$\mu =a+bd_{50}$	0.174	−0.0007	0.395	0.126
$\mu =a+bd_{60}$	0.134	−0.0004	0.395	0.126
$\mu =a+bd_{90}$	0.111	−0.0001	0.392	0.128
$\tau =a+bd_{10}$	2115.179	−30.948	0.934	0.831
$\tau =a+bd_{30}$	3581.505	−38.847	0.935	0.831
$\tau =a+bd_{50}$	−340.291	5.734	0.934	0.830
$\tau =a+bd_{60}$	−46.059	2.770	0.934	0.830
$\tau =a+bd_{90}$	118.957	0.915	0.936	0.836

## Data Availability

Not applicable.

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
