# Peer review of "Time-Dependent Rheological Properties of Cemented Aeolian Sand-Fly Ash Backfill Vary with Particles Size and Plasticizer"

_materials, 2023, doi:10.3390/ma16155295_

Round 1

Reviewer 1 Report

Respected Authors

I appreciate the time that you devoted to compose your study. I like very much laboratory based contributions with prospects to practical outcomes. There is only one issue that prevents me so far from direct accepting of your work. It is a very poorly written introductory part of your study.

Firstly, you should avoid cluster citations like [1-8], [9-13], [17-22], [23-25] and [26-30]. Every cited paper deserves a cautious introduction to prove their importance and relevance for your study. Especially, when most of listed papers are not referenced in further sections of your work.

Big part of your references are domestic (Chinese) studies composing almost 40% of references your list. Yes, it is truth that Chinese authors published numerous studies on mining technologies but the same problem with backfill mixture transportation appear in the entire region of south-east Asia and generally in all the countries with deep mining areas. I appreciate that you found numerous works of M. Fall from Canada and B. Ercikdi from Turkey who are very famous for their work. I only pity that you neglected recent works of Russian authors e.g. DOI: 10.3390/min11070739 and DOI: 10.1016/j.trpro.2021.09.019 and especially DOI: 10.1016/j.matpr.2020.10.139 where the Authors made very similar studies to your current contribution (The rheological characteristics of the backfill mixture were obtained experimentally: viscosity; ultimate shear stress for the time up to 30 minutes). None of the papers listed above is mine so there is no personal gain behind my suggestions.

If you want to provide a valuable “State of the Art report” in your introductory part, you should provide more international point of view. That would help to attract international Readers and widen the citing potential of your study. Please consider my opinion and also make a wider search in MDPI, Web of Science and/or Scopus.

Sincerely

Reviewer 3 Report

Reviewer’s comments:

# Authors have successfully determined the time-dependent rheological properties of fresh CAFB treated with the Master 480 Glenium plasticizer and cured under different ratio of aeolian sand/coal gangue.

However;  according to my point of view if the following things may strengthen the quality of the paper.

# Authors should   mention or calculate the density of slurry as it plays important role in shear stress and strain.

# Authors have performed experiments with plasticizer dosages of 0, 0.05, and 0.1% of the total CAFB weight. The reputability of the experiments may checked further by increasing more percentage above 0.1%.

# For ensuring the stability structure of slurry, author may use / or mention another comparative experimental techniques.

Round 2

Reviewer 1 Report

Respected Authors

I have no further comments.

I accepted current version of your study.

Sincerely

Author Response

Thank you for your valuable suggestion.

Reviewer 2 Report

Many of the concerns exposed in my review have not been taken into consideration and I sincerely do not understand why part of the text in the revised document in the "introduction" and "discussion" sections are highlighted in yellow, where except for the form in which the citations are expressed, nothing has changed. I leave the final decision on the publication of the final document to the Editor.
All the best.

Author Response

Dear reviewers, you have raised many questions in your review of the manuscript, and some of the revised text is highlighted in yellow, thank you for your valuable comments.